# Influence of the Active Layer Thickness of Permafrost in Eastern Siberia on the River Discharge of Nutrients into the Arctic Ocean

Olga I. Gabysheva [1], Viktor A. Gabyshev [1] and Sophia Barinova [2,*]

1   Institute for Biological Problems of Cryolithozone Siberian Branch of Russian Academy of Science (IBPC SD RAS), 677980 Yakutsk, Russia; g89248693006@yandex.ru (O.I.G.); v.a.gabyshev@yandex.ru (V.A.G.)
2   Institute of Evolution, University of Haifa, Mount Carmel, 199 Abba Khoushi Ave., Haifa 3498838, Israel
*   Correspondence: sophia@evo.haifa.ac.il

**Abstract:** Large rivers are important links between continents and oceans for material flows that have a global impact on marine biogeochemistry. Processes in the catchment areas of large rivers can affect the flow of solutes into the global ocean. The goal was to determine how the concentration of individual components of nutrients in the rivers of Eastern Siberia changes depending on the active layer thickness of the permafrost (ALT) and to elucidate whether the ALT is a factor that can control nutrient flux to the Arctic Ocean. The method of canonical correlation analysis was applied to the data on the concentration of nutrients in the 12 largest rivers of Eastern Siberia and the active layer thickness in their catchments. We found that the concentration of nutrients such as ammonium ion ($NH_4$) and total phosphorus ($P_{total}$) in river waters is higher in catchments with a deeper active layer. The waters of the mountain rivers in the south of the region (the Chara and Vitim rivers) are the richest in nutrients. Arctic rivers such as the Indigirka and Anabar were low in nutrients. The permeability of soils also affects the discharge of nutrients into rivers with surface runoff. We conclude that in the future, in the context of global climatic changes and the projected deepening of the active layer throughout the permafrost zone of the Northern Hemisphere, an increase in the supply of nutrients to the Arctic Ocean is possible.

**Keywords:** ammonium ion; total phosphorus; surface runoff; permafrost; rivers; East Siberian catchments

## 1. Introduction

Large rivers are important connections between continents and oceans for material flows that have a global impact on marine biogeochemistry. Processes in drainage basins of large rivers can affect the flow of dissolved substances from continents into the World Ocean, which can have serious consequences for such problems as coastal eutrophication, the development of hypoxic zones, and, as a result, for the biodiversity of aquatic biota. Therefore, the study of processes on the catchments of large rivers is important in the context of global changes [1].

The natural process of transfer of soluble biogeochemical components from soils to river waters occurs due to surface runoff. Under permafrost conditions, this process can be significantly influenced by such factors as the active layer thickness of permafrost (ALT). The territory of Eastern Siberia is characterized by an almost ubiquitous distribution of continuous permafrost to the north of the Vilyui River and discontinuous to the south. ALT in the study area ranges from 0.1–3.0 m and represents the uppermost permafrost horizon, subject to seasonal thawing in the warm season and freezing at subzero air temperatures.

In spring, groundwater runoff is limited to the topsoil, regardless of whether the area is covered by permafrost or seasonally frozen ground. Later, closer to the summer, low runoff, permafrost, or rather, that of the active layer thickness, has a strong land runoff effect on the

depth of penetration into the soil. The thinner active layer keeps the surface flow close to the soil surface, increasing the interaction of water with the upper soil horizon. The deeper active layer provides surface runoff through the underlying horizons of mineral-rich soil. It is obvious that such a difference in ground flow paths affects the chemical composition of waters due to exchange reactions between soil and water, which vary depending on the depth of the soil horizon.

There is uncertainty about how ALT can control nutrient inputs to Arctic rivers. Conflicting data were obtained when comparing the concentration of dissolved inorganic nitrogen (DIN) in the catchment areas of Alaska [2–5] and Western Siberia [6] with discontinuous permafrost. When comparing data on the concentration of dissolved silicate and phosphate in the cryolithozone in Alaska [2] and in Western Siberia [7], a direct relationship with ALT was established. Monitoring studies of the content of nitrates in the rivers of Alaska showed an increase in their concentration over time, which may be associated with the observed processes of degradation of permafrost [8].

It should also be noted that the processes of permafrost degradation in the catchments of the permafrost zone have a great potential to influence not only the concentration and transport of soluble substances by rivers, but also the transport of suspended solids, which may also contain nutrients [9]. These processes can also affect changes in river runoff [10] and local vegetation [11].

According to known estimates, in the catchment area of the Lena River in Eastern Siberia, as well as throughout Russia, an increase in ALT has been noted in recent decades [12,13]. The forecast of ALT dynamics made by Stendel and Christensen [14] shows the possibility of its increase by 30–40% for most areas of permafrost distribution in the Northern Hemisphere by 2100. Consequently, knowledge on the mechanisms of the ALT effect on the removal of soluble biogeochemical constituents from permafrost catchments can serve as a basis for predicting changes in the chemical composition of river waters, as well as for assessing the rate of inflow of dissolved substances into the Arctic Ocean.

Thus, the relevance of studying the impact of ALT in the catchments of Eastern Siberia on the concentration of nutrients in river waters is obvious. We hypothesize that the removal of nutrients into the Arctic Ocean is associated with the thickness of the active permafrost layer in the catchments of Eastern Siberia rivers and with the dispersal of rocks.

The region studied by us is characterized by continuous permafrost, in contrast to other regions where similar studies were carried out (Alaska, West Siberia, Canada), where permafrost is discontinuous. Obviously, studies carried out in regions with extensive discontinuous permafrost, where permafrost covers 50–90% of the watersheds, or sporadic permafrost, where permafrost cover is less than 50%, are less reliable than in Eastern Siberia, where permafrost covers 100%. In addition, poor economic development and a small population of the studied region make it possible for us to exclude the impact of anthropogenic factors on the analysis and focus only on ALT as the natural factor.

The aim of this work was to determine how the concentration of specific constituents of nutrients in the rivers of Eastern Siberia changes depending on the active layer thickness of permafrost and to find out if ALT is the factor that can control nutrient flux to the Arctic Ocean.

## 2. Materials and Methods

The study is based on observations on 12 large rivers of Eastern Siberia: Lena, Vilyuy, Kolyma, Aldan, Olenyok, Vitim, Indigirka, Amga, Olyokma, Anabar, Yana, and Chara (Figure 1). The study area is located between 56°13'and 73°10' N, and in longitude extends from 106°53' to 160°58' E. The studied rivers flow in the zone of northern taiga and tundra; the most typical landscapes of the investigated catchments are shown in Figure 2. The largest of the studied rivers, the Lena, is one of the 15 largest rivers in the world in terms of length, basin area, and average annual water discharge. The main hydrological and morphometric indicators of the studied rivers are presented in Table 1. In the south of the region, in the mountains, the rivers are characterized by the highest mean stream gradient

(Chara, Olyokma). For the rivers of the Arctic part of the region, located closer to the Arctic Ocean, the value of this indicator is the lowest (Olenyok, Yana). Water sampling was carried out at the summer runoff (June-August) during 2007–2011. A total of 303 water samples were taken from the near-surface horizon (0–0.3 m) in the coastal zone and along the river bed.

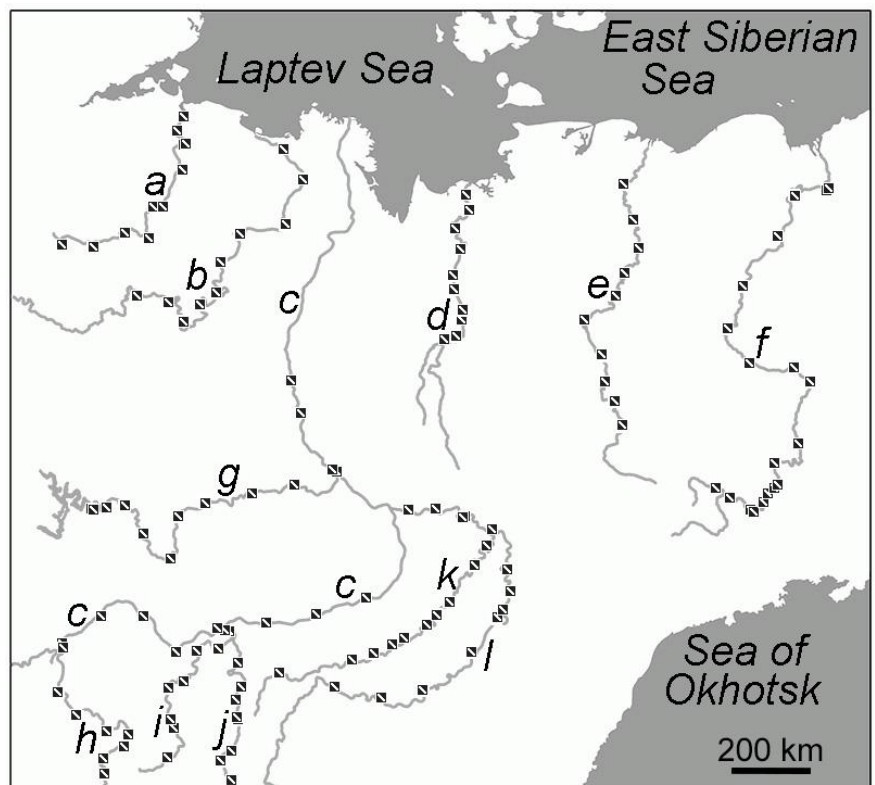

**Figure 1.** Sketch map of study area and sampling points. Rivers: (**a**) Anabar, (**b**) Olenyok, (**c**) Lena, (**d**) Yana, (**e**) Indigirka, (**f**) Kolyma, (**g**) Vilyuy, (**h**) Vitim, (**i**) Chara, (**j**) Olyokma, (**k**) Amga, and (**l**) Aldan.

The preservation and storage of samples was carried out in accordance with generally accepted methods [15]. The concentration of nutrients was assessed photometrically using an SF-26 spectrophotometer: ammonium ions with Nessler's reagent, nitrite ions with Griss' reagent, nitrate ions with sodium salicylate, phosphate ions and silica by the method of molybdenum blue, total phosphorus by persulfate oxidation, and total iron using sulfosalicylic acid. We published the information on the concentration of nutrients for each sampling point earlier [16].

The generated dataset also included information on the geographic coordinates of the sampling points, information on the ALT, and the dispersion of parent rock material. ALT data (minimum, average, and maximum active layer thickness, m) were obtained from Beer et al. [17]. In this work, the ALT map is compiled on the basis of the Permafrost-landscape map of the Yakutian ASSR [18]. The development of the special content of this map was carried out based on the materials of aerospace research and special field studies published in the works of the leading institutes of the Siberian Branch of the USSR Academy of Sciences [19]. The gridded dataset from which we extracted ALT information using the ArcGIS software package according to the coordinates of our observation points is the result of digitizing this original map into multiple GIS layers at a scale of 1:2,500,000, with a final resolution of up to 0.5 degrees. The described dataset is available in the PANGEA [20] repository as a NetCDF file. The information on the dispersion of parent rock material was taken from the National Atlas of Soils of the Russian Federation [21], where this variable is estimated in ranks differing by fractions and grain size of particles (Table 2). "Dispersion

of parent rock material" is a grouping variable that combines five clusters of sampling points that takes the following values: 1–1–2 ranks; 2–2–3 ranks; 3–3–4 ranks; 4–4–5 ranks; 5–5–6 ranks.

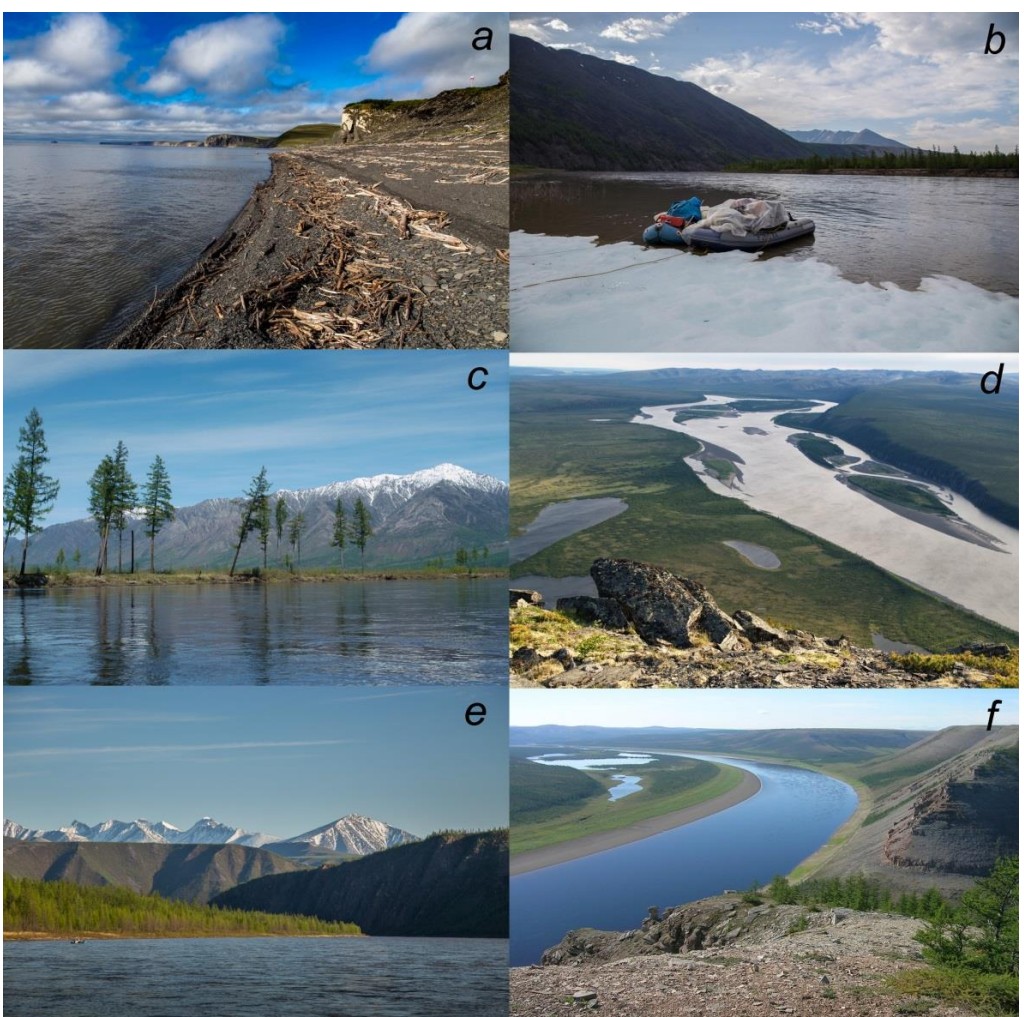

**Figure 2.** Typical landscapes of the studied rivers: (**a**) lower reach of the Lena River in tundra, (**b**) Kolyma River and sheet of frazil on the river bank, (**c**) Chara River, one of the mountain rivers on the south of the region, (**d**) Yana River, (**e**) Indigirka River, (**f**) Olenyok River.

**Table 1.** Main characteristics of the rivers and their catchment areas.

| Rivers | Stream Length, km | Watershed Area, km$^2$ | Mean Stream Gradient, ‰ | Average Annual Water Discharge, m$^3$ s$^{-1}$ |
|---|---|---|---|---|
| Lena | 4270 | 2,420,000 | 0.33 | 16,350 |
| Vilyuy | 2650 | 454,000 | 0.42 | 1480 |
| Kolyma | 2600 | 665,000 | 0.25 | 2250 |
| Aldan | 2273 | 729,000 | 0.55 | 5246 |
| Olenyok | 2270 | 219,000 | 0.22 | 1210 |
| Vitim | 1916 | 227,200 | 0.54 | 1520 |
| Indigirka | 1900 | 362,000 | 0.46 | 1600 |
| Amga | 1360 | 75,000 | 0.47 | 178 |
| Olyokma | 1310 | 201,200 | 0.98 | 1950 |
| Anabar | 939 | 100,000 | 0.48 | 498 |
| Yana | 872 | 238,000 | 0.15 | 1021 |
| Chara | 851 | 87,600 | 1.09 | 639 |

**Table 2.** Grading the dispersion of parent rock material, ranks (according to Alyabina [21]).

| Rock and Fraction | Grain Size, mm | Rank |
|---|---|---|
| Massive rock | - | 1 |
| Semi-rock | - | 2–3 |
| Large-sized fraction | >2.0 | 3 |
| Sand | 2–0.05 | 4 |
| Loess | 0.05−0.005 | 5 |
| Clay | <0.005 | 6 |

Canonical Correlation Analysis (CCA) reveals the relationship between two sets of quantitative variables [22]. Using a multivariate model to analyze canonical values also allows data to be distinguished between multiple clusters of a grouping variable. We used the CCA to investigate the relationship between sets of quantitative variables describing ALT and nutrient concentrations in river waters. We also used it to split the data between the 12 studied rivers and the 5 clusters of the "Dispersion of parent rock material" grouping variable. The analysis algorithm includes the input into the analysis of two sets of variables and the subsequent determination of the canonical correlation (R) between them and the level of significance ($p$). At the next stage of the analysis, the maximum modules of the normalized canonical coefficients for the canonical axes of each set of quantitative variables were estimated. At the final stage, the scatterplot allows determining how the observations categorized by groups are located relative to its axes. A 5% significance level was used to test the statistical hypothesis. Statistical analysis was performed using the Statistica 10 software package.

## 3. Results

To search for a canonical correlation, two sets of quantitative variables were organized. The analysis included the set of seven variables characterizing the concentration of nutrients in the studied rivers ($NH_4$, mg $L^{-1}$; $NO_2$, µg $L^{-1}$; $NO_3$, mg $L^{-1}$; $PO_4$, µg $L^{-1}$; $P_{total}$, µg $L^{-1}$; Si, mg $L^{-1}$; $Fe_{total}$, mg $L^{-1}$) (hereinafter set NUTRIENTS) paired with a set of three variables of ALT (minimal, mean, and maximal active layer thickness, m). The strength of association between these two canonical variates was measured by the adjusted canonical correlation coefficient, calculated for the first canonical root, which was R = 0.41 with a significance level of $p$ <0.0001. The suitability of the obtained result for analysis is confirmed by an additional check of significance using four criteria (Table 3).

**Table 3.** Multivariate statistics and F approximations for canonical correlations between two sets of variables: ALT (3 variables) and NUTRIENTS (7 variables).

| Statistic | Value | F Value | Pr > F |
|---|---|---|---|
| Wilks' Lambda | 0.72 | 4.91 | <0.0001 |
| Pillai's Trace | 0.31 | 4.79 | <0.0001 |
| Hotelling-Lawley Trace | 0.36 | 5.01 | <0.0001 |
| Roy's Greatest Root | 0.23 | 9.80 | <0.0001 |

Standardized canonical coefficients are considered for a more detailed comparative analysis. The first canonical axes are of interest for the analysis since they have the maximum correlations with each other and are the most informative. Table 4 shows the correlation coefficients between the active layer thickness variables and the ALT1 canonical axis, ranked in descending order of their magnitude. These coefficients are dimensionless, so they are suitable for comparison with each other. The highest values of the analyzed canonical axis had variable "maximal active layer thickness, m"; other variables had minor values.

**Table 4.** Correlations between the ALT1 and their canonical variables.

| Name of Variable | Standardized Canonical Coefficients for the ALT1 |
| --- | --- |
| Maximal active layer thickness, m | −1.3190 |
| Mean active layer thickness, m | −0.8880 |
| Minimal active layer thickness, m | −0.7877 |

The highest values of standardized canonical coefficients among examined nutrients were the following variables: "$NH_4$, mg $L^{-1}$" and "$P_{total}$, µg $L^{-1}$" (Table 5).

**Table 5.** Correlations between the NUTRIENTS1 and their canonical variables.

| Name of Variable | Standardized Canonical Coefficients for the NUTRIENTS1 |
| --- | --- |
| $NH_4$, mg $L^{-1}$ | −0.7165 |
| $P_{total}$, µg $L^{-1}$ | −0.6168 |
| $NO_2$, µg $L^{-1}$ | −0.3259 |
| $Fe_{total}$, mg $L^{-1}$ | 0.2050 |
| $NO_3$, mg $L^{-1}$ | −0.1635 |
| Si, mg $L^{-1}$ | −0.1537 |
| $PO_4$, µg $L^{-1}$ | −0.0189 |

The relationship between ALT on the catchment and the concentration of nutrients in river waters was statistically significant. Moreover, the deeper the active layer in a catchment, the higher the concentration of nutrients in the rivers, mainly ammonium ion and total phosphorus.

Scatterplots of sampling points categorized by investigated rivers along the first two canonical axes allowed us to characterize the mountain rivers of the southern region (rivers Chara and Vitim) as the richest in nutrients (Figure 3). The waters of such arctic rivers as Indigirka and Anabar were characterized by the minimum content of nutrients. The lowest nutrients were found in the Vilyuy River. Scattering of sampling points along the *x*-axis (ALT1) shows that a thinner seasonally thawing layer of permafrost characterizes the catchments of such arctic rivers as Indigirka, Olenyok, Anabar, and Yana. On the contrary, sampling points are shifted towards a deeper active layer for the catchments of rivers in the south of the region (Chara, Olyokma, and Vitim). Scattering of sampling points along the *x*-axis (ALT1) was more even for extended rivers crossing significant distances along the meridian (Lena, Kolyma), which indicates a large variation in the ALT at different sections of these rivers (Figure 3).

Obviously, the effect of ALT on the export soluble biogeochemical components from soils to rivers can be largely corrected due to the permeability of soils, which is related to the soil grain size. It is known that soil grain size is derived from parent rock material [23]. Therefore, we used the available data on the degree of dispersion of the parent rock material [21] to assess how the water permeability of soils can be superimposed on the regulatory function of ALT during the exchange of soluble biogeochemical constituents between water and soil.

Here, we show scatterplots of the sampling points divided into five clusters of the "dispersion of parent rock material" grouping variable along the first two canonical axes derived before (Figure 3). The number of sampling points located at the top of the scatterplot is reduced from cluster 1, characterized by low ranks of dispersion, to cluster 5, which combines high dispersive parent rock material. This can be clearly seen by drawing a notional line through mark 2 on the *Y*-axis of NUTRIENTS1: the largest number of sampling points above this line belongs to catchment sections with low ranks of dispersion of parent rock material (Figure 4). Consequently, in those catchments areas where finely dispersed material of the parent rock is present, the concentration of nutrients in the rivers is lower.

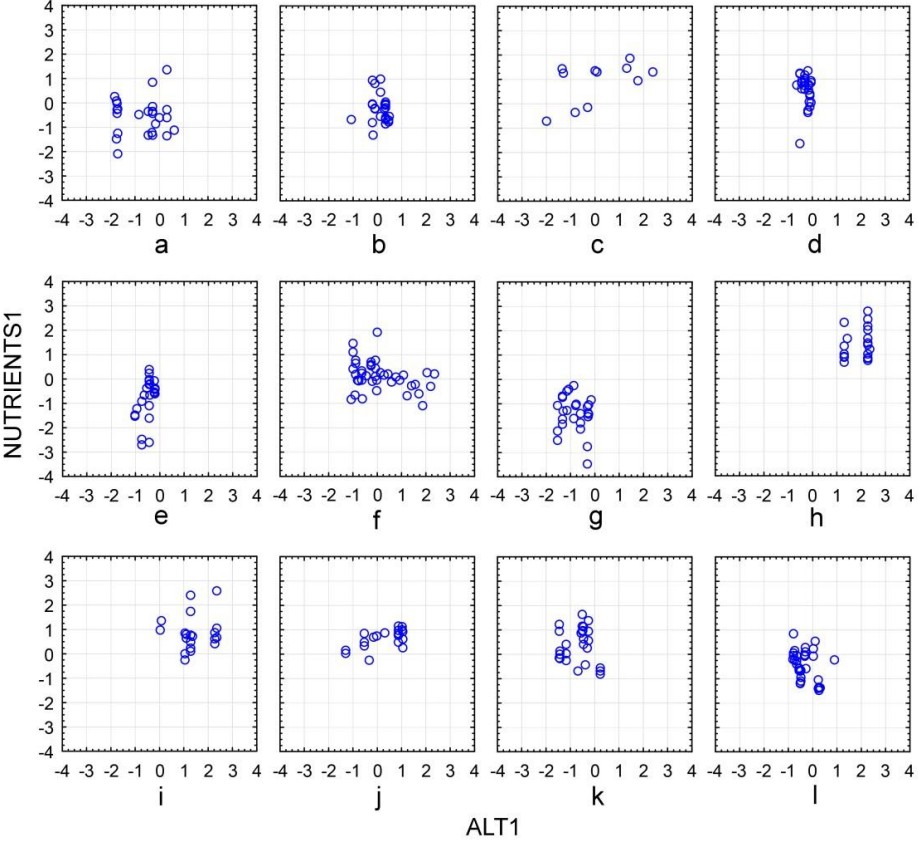

**Figure 3.** Scatterplot of sampling points along the first two canonical axes (NUTRIENTS1 against ALT1) categorized by investigated rivers. Rivers: (**a**) Anabar, (**b**) Olenyok, (**c**) Lena, (**d**) Yana, (**e**) Indigirka, (**f**) Kolyma, (**g**) Vilyuy, (**h**) Vitim, (**i**) Chara, (**j**) Olyokma, (**k**) Amga, and (**l**) Aldan.

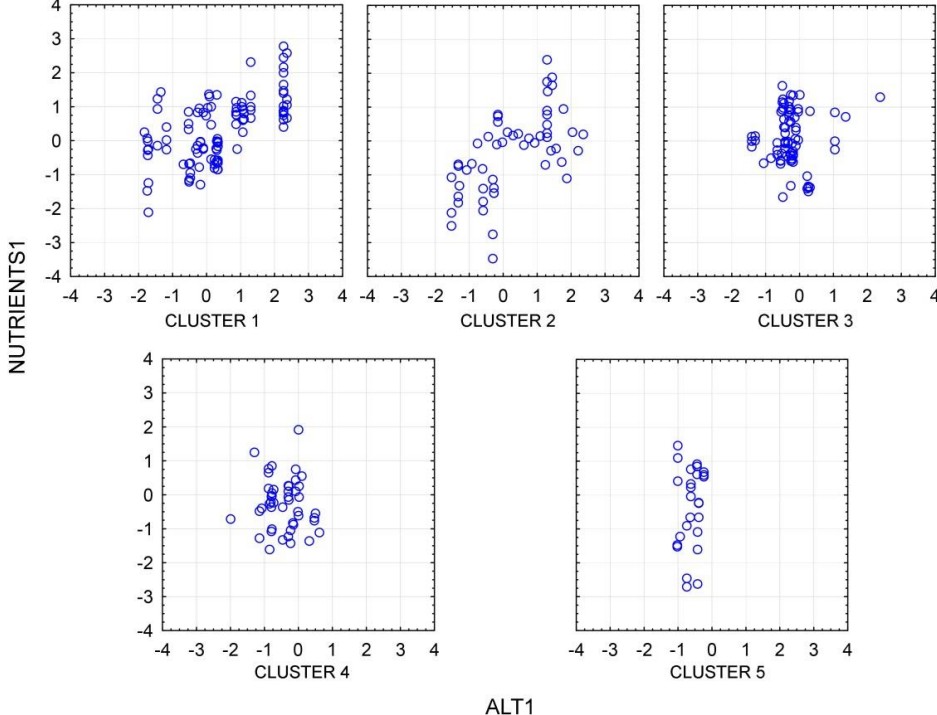

**Figure 4.** Scatterplot of sampling points along the first two canonical axes (NUTRIENTS1 against ALT1) categorized by five clusters of the grouping variable "dispersion of parent rock material".

## 4. Discussion

The results of the analysis confirm that for the territory of Eastern Siberia, there is a relationship between the ALT of permafrost on the catchment and concentration of nutrients in river waters. This dependence is not so pronounced; this is obviously because the chemical composition of river waters is formed in a complex system, where many different factors operate. Nevertheless, a factor such as ALT influences the export of nutrients from catchments into the rivers of the studied region. This effect is because the ALT of permafrost regulates the depth of flow paths of water (surface runoff) and the exchange of soluble biogeochemical constituents between water and soil. The deeper the ALT and, as a consequence, the deeper the depth of flow paths, the higher the nutrient concentrations in the rivers of the studied region.

ALT in the conditions of Eastern Siberia has the greatest influence on the export to rivers of nutrients such as ammonium ion and total phosphorus. In Alaska's discontinuous permafrost catchments, a comparison was made between data collected from adjacent permafrost and non-permafrost areas. The results of the work led the researchers to the conclusion that deeper flow paths in areas devoid of permafrost contribute to a more significant export of inorganic nitrogen [3,5]. This hypothesis is also supported by a study comparing the concentration of inorganic nitrogen in river waters of Alaska between 63°12′ and 68°5′ N, according to which its content is, on average, higher in the south of the region with discontinuous permafrost, in comparison with the high latitude section with continuous permafrost [3].

Anomalously high concentrations of soluble reactive phosphorus found in a stream on the North Slope of Alaska have also been associated with permafrost degradation [24]. Since mineral weathering is the primary source of phosphate in soil, deeper flow paths through previously frozen mineral soils, such as ALT deepening, lead to an increase in the concentration of this element [10].

ALT loses nutrient deposits due to their constant removal with overland flow. Meanwhile, significant reserves of nutrients are sequestered within the near-surface permafrost. Thus, the results of a comparative study of permafrost and active layer soils in Alaska showed that the concentration of exchangeable phosphorus is greater in permafrost soils [15]. In this regard, the increase in ALT noted in recent decades [12,13] may lead to an increase in the concentration of nutrients in the rivers of the cryolithozone. This is confirmed by the data of long-term observations in the upper Kuparuk River basin in Alaska's North Slope, where a significant increase in the concentration and export of inorganic nitrogen was noted in the 1990s [8], which researchers associated with the processes of permafrost degradation in this region.

We did not note the effect of ALT of catchments on the increase in silicate concentration in the studied rivers. However, this pattern was found when comparing data from different sites with discontinuous permafrost in Alaska [2,25].

Opposite results were obtained in studies in Western Siberia, where a comparison of 96 rivers over a wide latitudinal range found no differences in nitrate concentrations linked to variations in permafrost coverage [6]. This result is explained by the weak water permeability of soils in the region due to the extensive peatlands that characterize Western Siberia [10].

Our results also showed that soil grain size and its associated weak water permeability overlap with ALT's regulatory function and inhibit the export of soluble biogeochemical constituents by surface runoff.

It should be noted that, apart from natural factors such as ALT, factors associated with human activities (industry, agriculture, household drains) also determine the flow of nutrients into rivers in economically developed regions of the world. However, the area to which our study belongs is one of the most sparsely populated and poorly developed regions of the world. Therefore, on the territory of Yakutia, the area of which is 3.08 million $km^2$, only 0.98 million people live there, and more than a third is concentrated in one settlement in the city of Yakutsk (347.2 thousand people). The population density averages 0.32 ppl/$km^2$

in the most populated areas in the center and south of the region up to 2.6 ppl/km$^2$, and in the Arctic part of the region, there is only 0.04 ppl/km$^2$. In this regard, human activities and related factors that can lead to an increase in the concentration of nutrients in the river runoff are very limited in this region. This allows us to exclude the anthropogenic factor from our analysis.

It is important to note that some of the nutrients may not reach the Arctic Ocean and pass into bottom sediments, combining with particles of suspended solids and thus forming river sediments [9]. However, our results confirm the hypothesis that the export of nutrients to the Arctic Ocean largely depends on ALT variations on catchments of the cryolithozone, which controls the dominant flow paths of water. In recent decades, for the entire territory underlain by permafrost in the Northern Hemisphere, there has been an increase in ALT [12]. There are forecasts of ALT increase in the current century [14]. In this regard, it is obvious that the rate of input of nutrients into the Arctic Ocean from the territory of Eastern Siberia may increase in the future.

## 5. Conclusions

It was found that the concentration of ammonium ion and total phosphorus is associated with the depth of seasonal thawing of permafrost based on the analysis of the original data of the nutrient content and available ALT data from the studied catchments of the 12 largest rivers in Eastern Siberia. It was shown that with an increase in ALT and, consequently, an increase in the depth of flow paths of water (surface runoff), the concentration of these nutrients increases. Rivers such as the Chara and Vitim, which flow in the south of the region, where ALT is deeper, are characterized as the richest in nutrients. The minimum concentration of nutrients was noted for such rivers as Indigirka, Anabar, and Vilyuy, flowing in the north of the studied region with a thinner active layer. Our results are consistent with the data of studies carried out in the catchments of Alaska.

The relationship between the ALT of permafrost in the catchment area and the content of nutrients in river waters, which we identified, is not so pronounced. This is due to the objective limitations of this study associated with the fact that the chemical composition of river waters is formed in a complex system, where many different factors operate. Modeling such systems has a known level of uncertainty. Nevertheless, it is obvious that a factor such as ALT affects the supply of nutrients to the rivers of the studied region.

Thus, the future export of nutrients to the Arctic Ocean will likely be controlled by changes in spatial patterns of the ALT influencing the dominant flow paths of water in Arctic river catchments. The ALT deepening noted in recent decades in the cryolithozone of the Northern Hemisphere may continue in the current century. Therefore, an increase in the supply of nutrients to the Arctic Ocean in the future is possible.

A different degree of water permeability of soils is superimposed on the regulating function of ALT. In catchments with a higher rank of parent rock material dispersion and with weak water permeability of soils, the export of soluble biogeochemical constituents by surface runoff is hindered, and the concentration of nutrients in rivers is lower.

**Author Contributions:** O.I.G. designed the work, performed chemical-analytical analysis, writing, and revision of the manuscript. V.A.G. contributed to the design of the work and revision of the manuscript, collecting field data, statistical analysis, funding acquisition. S.B. collaborated in the revision of the manuscript. All authors have read and agreed to the published version of the manuscript.

**Funding:** The research was carried out within the state assignment of the Ministry of Science and Higher Education of the Russian Federation (theme No. 0297-2021-0026, reg. No. AAAA-A21-121012190036-6).

**Institutional Review Board Statement:** Not applicable.

**Informed Consent Statement:** Not applicable.

**Data Availability Statement:** Not applicable.

**Acknowledgments:** This work was partly supported by the Israeli Ministry of Aliyah and Integration.

**Conflicts of Interest:** The authors declare no conflict of interest.

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
