# Peer review of "Influence of the Active Layer Thickness of Permafrost in Eastern Siberia on the River Discharge of Nutrients into the Arctic Ocean"

_water, doi:10.3390/w14010084_

Round 1

Reviewer 1 Report

  1. A short brief of canonical analysis is needed in materials and methods section.
  2. Scatterplot diagram figure 2 and 3- please show the diagram in square box having equal scale -4 to 4. It would provide clear picture for good correlation, underestimation and overestimation.
  3. If possible, please provide the ALT vs Nutrients along the river. How the correlation between ALT vs Nutrient varies from headwaters reach to downstream.
  4. Line 218-220, Figure 2 and 3 are not enough to draw the conclusion.
  5. The purpose of this study was mentioned as "to determine the specific constituents of nutrients that are affected by the ALT of permafrost, as well as to assess how the concentration of these nutrients in the rivers of East Siberia changes depending on ALT and the role of this process in the removal of nutrients into the Arctic ocean". Assessment of the physics and the role of this process in the removal of nutrients into the Arctic ocean is lacking.

Author Response

Thank you the Reviewer 1 for comments that improved my ms. Please find below the responses to each comments

With best regards,

Prof Sophia Barinova

Reviewer 1

  1. Reviewer: A short brief of canonical analysis is needed in materials and methods section.

Authors: A short brief of canonical analysis added in methods section.

  1. Reviewer: Scatterplot diagram figure 2 and 3- please show the diagram in square box having equal scale -4 to 4. It would provide clear picture for good correlation, underestimation and overestimation.

Authors: Scatterplot diagram figure 2 and 3 redesigned.

  1. Reviewer: If possible, please provide the ALT vs Nutrients along the river. How the correlation between ALT vs Nutrient varies from headwaters reach to downstream.

 Authors: To display 7 indicators of nutrients and 3 indicators of ALT for 303 sample points, it will be necessary to present a rather cumbersome table in the article. But this will not give an understanding of the patterns in their relationship. Not all studied rivers flow from south to north, for example, the Vilyui River and the middle Lena flow from west to east. And ALT does not everywhere increase from north to south, for example, in the mountains, the distribution of ALT is more complicated. To find these patterns, the CCA statistical method was applied. A simple correlation of these parameters will not give an understanding of the relationship between them.

  1. Reviewer: Line 218-220, Figure 2 and 3 are not enough to draw the conclusion.

 Authors: The conclusion is based on analysis of scatterplot of sampling points categorized by 5 clusters of grouping variable “dispersion of parent rock material” along the first two canonical axes (presented in Figure 4).

  1. Reviewer: The purpose of this study was mentioned as "to determine the specific constituents of nutrients that are affected by the ALT of permafrost, as well as to assess how the concentration of these nutrients in the rivers of Eastern Siberia changes depending on ALT and the role of this process in the removal of nutrients into the Arctic ocean". Assessment of the physics and the role of this process in the removal of nutrients into the Arctic Ocean is lacking.

 Authors: Made appropriate adjustments to the formulation of the research objective.

Reviewer 2 Report

This manuscript concerns a research paper on the relationship between the permafrost thickness and the varying concentration of specific constituents of nutrients in the rivers. The research herein presented is certainly within the scope of Water.

The manuscript is in good shape and I just leave the authors with a list of short comments. My current decision is minor revision. I will be happy to review a revised version of the manuscript.

Comments

  1. First paragraph. Physical processes in the catchment areas of large rivers do not only have an impact on the transport of solutes but also on the transport of sediments (which can convey nutrients attached to the sediment particles) [1] and on the spatiotemporal dynamics of water volumes and local vegetation [2]. This should be acknowledged and clarified.
  2. Section 2. A table with the characteristics of the river catchment areas should be included. This table needs to include the area of each catchment, the river length, the mean slope, etc. This is important for comparison with other studies and also to assess the effectiveness of the methodology herein outlined given the spatial scale of the river catchment.
  3. Figure 2 is a bit blurry. Please, increase the quality image.
  4. Discussion section. The results outlined in the manuscript confirm the relationship between the permafrost and the concentration of nutrients in river waters. Nonetheless, I think this finding should be further explored and linked in the future with the transport of sediments. In spring, overland flows transport huge amounts of sediments…and nutrients which are attached to the sediment particles. I think the observations of sediment hysteresis [3] could help to improve the understanding on the supply/storage of nutrients in these rivers.

Bibliography

[1] Intraseasonal‐to‐Interannual Analysis of Discharge and Suspended Sediment Concentration Time‐Series of the Upper Changjiang (Yangtze River). Juez. C. et al. WATER RESOURCES RESEARCH. 2021.

[2] Improved Understanding of the Link Between Catchment-Scale Vegetation Accessible Storage and Satellite-Derived Soil Water Index. L. J. E. Bouaziz et al. WATER RESOURCES RESEARCH. 2021.

[3] The origin of fine sediment determines the observations of suspended sediment fluxes under unsteady flow conditions. Juez, C. et al. WATER RESOURCES RESEARCH. 2018.

Author Response

Thank you the Reviewer 2 for comments that improved my ms. Please find below the responses to each comments

With best regards,

Prof Sophia Barinova

Reviewer 2

  1. Reviewer: First paragraph. Physical processes in the catchment areas of large rivers do not only have an impact on the transport of solutes but also on the transport of sediments (which can convey nutrients attached to the sediment particles) [1] and on the spatiotemporal dynamics of water volumes and local vegetation [2]. This should be acknowledged and clarified.

Authors: This is an important note, the corresponding explanations have been included in the text.

  1. Reviewer: Section 2. A table with the characteristics of the river catchment areas should be included. This table needs to include the area of each catchment, the river length, the mean slope, etc. This is important for comparison with other studies and also to assess the effectiveness of the methodology herein outlined given the spatial scale of the river catchment.

Authors: We added the table with main characteristics of the rivers and their catchment areas

  1. Reviewer: Figure 2 is a bit blurry. Please, increase the quality image.

Authors: We redesigned Figure 2(3) and 3(4) and the resolution of the images was raised up to 650 dpi

  1. Reviewer: Discussion section. The results outlined in the manuscript confirm the relationship between the permafrost and the concentration of nutrients in river waters. Nonetheless, I think this finding should be further explored and linked in the future with the transport of sediments. In spring, overland flows transport huge amounts of sediments…and nutrients which are attached to the sediment particles. I think the observations of sediment hysteresis [3] could help to improve the understanding on the supply/storage of nutrients in these rivers.

Authors: The authors are grateful to the Reviewer for this promising proposal concerning the development of this topic in the future. Studying the effect of the permafrost degradation process on the transport of suspended solids by rivers, the formation of river sediments and bottom sediments is an important task for researchers working in the permafrost zone.

Reviewer 3 Report

The results obtained look quite expected. With an increase in the volume of rocks involved in active drainage, including chemical denudation, the runoff of nutrients also increases. Here, I do not see a significant conceptual breakthrough in science, although the authors confirmed the above pattern using the example of the vast region of Eurasia covered by permafrost.

  1. My concern with the study's findings is that the authors associate spatial nutrient changes in rivers only with spatial variations in depth of the active layer of permafrost. But what about human activities? In the more southern regions of Eastern Siberia, as the annual average temperature and the duration of the warm season of the year increase, the conditions for a denser settlement of the population (primarily along the local rivers) and its economy (industrial enterprises, domestic wastewater, cultivated land, etc.) become better. However, in this direction, the depth of the active layer of permafrost also increases. For example, the primary human-induced sources of phosphorus in rivers are household and industrial wastewater (primarily synthetic detergents), runoff from farmland when using phosphorus fertilizers, runoff from livestock farms. The presence of ammonium ions in concentrations exceeding background values may result from the poor-quality operation of water treatment facilities, livestock farms, accumulations of manure, nitrogen fertilizers, settlements, and tourist camping. There is no analysis of this in the manuscript.  I am stunned by this fact. Analysis of the influence of one factor should be carried out while excluding the impact of other factors.
  2. There is no information in the manuscript about the limitations and uncertainties of the study.
  3. Lines 101-103. The technique for determining the depth characteristics of the active layer of permafrost is revealed superficially. Much is not clear. Please pay more attention to this.
  4. Not the entire mass of nutrients mobilized in the basins of the studied rivers reaches the Arctic Ocean. Some of them pass into river alluvium with suspended particles of river waters. The larger a river basin, the greater these losses during the formation of various sediments, especially in the coastal lowlands of the northern part of the study region. This issue should also be discussed in some way in the manuscript.
  5. The English language of the manuscript needs improvement.

In addition:

  1. I would correct the title of the manuscript as follows: “Influence of the depth of the active layer of permafrost in Eastern Siberia on the river discharge of nutrients into the Arctic Ocean”.
  2. Figure 1 needs a linear scale.
  3. "Horton overland flow"? The meaning does not change in any way if you write "surface runoff".

Author Response

Thank you the Reviewer 3 for comments that improved my ms. Please find below the responses to each comments

With best regards,

Prof Sophia Barinova

Reviewer 3

  1. Reviewer: My concern with the study's findings is that the authors associate spatial nutrient changes in rivers only with spatial variations in depth of the active layer of permafrost. But what about human activities? In the more southern regions of Eastern Siberia, as the annual average temperature and the duration of the warm season of the year increase, the conditions for a denser settlement of the population (primarily along the local rivers) and its economy (industrial enterprises, domestic wastewater, cultivated land, etc.) become better. However, in this direction, the depth of the active layer of permafrost also increases. For example, the primary human-induced sources of phosphorus in rivers are household and industrial wastewater (primarily synthetic detergents), runoff from farmland when using phosphorus fertilizers, runoff from livestock farms. The presence of ammonium ions in concentrations exceeding background values may result from the poor-quality operation of water treatment facilities, livestock farms, accumulations of manure, nitrogen fertilizers, settlements, and tourist camping. There is no analysis of this in the manuscript.  I am stunned by this fact. Analysis of the influence of one factor should be carried out while excluding the impact of other factors.

Authors: The authors are grateful to the Reviewer for focusing attention on the problem of anthropogenic impact. Our mistake was that we did not make an appropriate explanation in the text about why we neglected the anthropogenic impact of in our analysis. The fact is that the study area belongs to the most sparsely populated and poorly developed regions of the world. Thus, 0.98 million people live on the territory of Yakutia, and more than a third is concentrated in one settlement - the city of Yakutsk (347.2 thousand people). The population density averages 0.32 ppl/km2, in the most populated areas in the center and south of the region - up to 2.6 ppl/km2, and in the Arctic part of the region - only 0.04 ppl/km2. In this regard, human activities and related factors that can lead to an increase in the concentration of nutrients in the river runoff are very limited in this region. A relevant clarification has been added to Section 3

  1. Reviewer: There is no information in the manuscript about the limitations and uncertainties of the study.

Authors: In Section 5, we have made an appropriate clarification regarding the existing limitations and uncertainties of the study. This will allow us to take a more balanced position in the conclusions of the work.

  1. Reviewer: Lines 101-103. The technique for determining the depth characteristics of the active layer of permafrost is revealed superficially. Much is not clear. Please pay more attention to this.

Authors: Thank you for an important clarification, this moment was overlooked by us. We have corrected this defect and added additional clarification to the text and the corresponding references to literary sources.

  1. Reviewer: Not the entire mass of nutrients mobilized in the basins of the studied rivers reaches the Arctic Ocean. Some of them pass into river alluvium with suspended particles of river waters. The larger a river basin, the greater these losses during the formation of various sediments, especially in the coastal lowlands of the northern part of the study region. This issue should also be discussed in some way in the manuscript.

Authors: We tried to briefly discuss the fact that some of the nutrients may not reach the Arctic Ocean and are deposited in bottom sediments, contributing to the formation of river sediments. Provided in the text the appropriate link to the publication on this issue. However, this moment does not negate the fact that ALT makes a significant contribution to the transfer of nutrients from the basin to the rivers and further to the Ocean.

  1. Reviewer: The English language of the manuscript needs improvement.

Authors: English was corrected by a native speaker.

additional notes:

  1. Reviewer: I would correct the title of the manuscript as follows: “Influence of the depth of the active layer of permafrost in Eastern Siberia on the river discharge of nutrients into the Arctic Ocean”.

Authors: The title of the article was changed in accordance with the comment of the Reviewer.

  1. Reviewer: Figure 1 needs a linear scale.

Authors: A scale bar has been added to Figure 1.

  1. Reviewer: "Horton overland flow"? The meaning does not change in any way if you write "surface runoff".

Authors: The term "Horton overland flow" has been changed to "surface runoff".

Reviewer 4 Report

Please see the file attached.

Author Response

Thank you the Reviewer 4 for comments that improved my ms. Please find below the responses to each comments

With best regards,

Prof Sophia Barinova

Reviewer 4

  1. Reviewer: [Introduction] The Authors well emphasize the relevance and the purpose of their study. However, could the Authors better highlight the novelties and advances in knowledge their study would provide in comparison to other literature studies on the same topic?

Authors: The authors are grateful to the Reviewer for constructive comments. Corresponding additions were made to the text of Section 1.

  1. Reviewer: [Materials and Methods] (i) This section lacks of hydrological and hydraulic characteristics of the (large) rivers under study. More in general I would provide a table in which the environmental conditions of the 12 large rivers under study are provided. For instance, for each river the Authors should specify: The drainage area, the percentage of the permafrost coverage; the average active layer thickness; typical values for the runoff (maybe characterized by seasons); typical values of the Froude number….. (ii) I don’t think it’s superfluous to add some comments on how the Active layer Thickness was estimated/measured for each river/catchment; (iii) It would be nice to add some emblematic photographs for the rivers under study.

Authors: A corresponding table with the main parameters of the rivers has been added. Also added clarification about ALT data source. Photos of the explored rivers have been added to the text of the article.

  1. Reviewer: [Results and Discussion] (i) Could the Authors better emphasize the general impact of their results? In other words, could the Authors discuss how their results can be extended in different contexts? The Authors do not consider dimensionless variables in their study and this leave some perplexity on the general impact of their findings; (ii) Could there be any effect of the temperature on nutrient release?

Authors: Of course, air temperature has a direct impact on ALT. ALT is higher in areas where summer temperatures are high. However, we do not directly take into account the effect of the air temperature factor in our study, since it acts on the release of nutrients from soil indirectly, through ALT. As for the soil temperature, the processes of nutrient transfer from the soil are completely blocked at soil temperatures below 0 °C. The removal of nutrients from the soil is possible only when the temperature of the soil rises above 0, when the soil opens up access for surface runoff.

Round 2

Reviewer 1 Report

As all comments are addressed in revised manuscript, the revised version would be acceptable.

Author Response

Responses to comments of Reviewer 1 Report 2

Comments and Suggestions for Authors

As all comments are addressed in revised manuscript, the revised version would be acceptable.

Response: Dear Reviewer 1, thank you for review our article and comments in respect of each the article text was corrected.

Reviewer 3 Report

My final comments:

  1. There is no need to use the same words and phrases in the Keywords as in the title of the manuscript. This reduces their effectiveness. Replace them with other valuable words or phrases from the text.
  2. Table 1. Discharge, m3 cm-1. What did you mean? I believe it is correct to write the following: Average annual water discharge, m3 s-1.
  3. Line 154. … the surface horizon (0–0.3 m)? the near-surface horizon (0–0.3 m)!

Author Response

Responses to comments of Reviewer 3 report 2

Dear Reviewer 3, thank you for review our article and comments in respect of each the article text was corrected

  1. There is no need to use the same words and phrases in the Keywords as in the title of the manuscript. This reduces their effectiveness. Replace them with other valuable words or phrases from the text.

Response: Changed keywords, eliminated duplication with the title of the article.

  1. Table 1. Discharge, m3 cm-1. What did you mean? I believe it is correct to write the following: Average annual water discharge, m3 s-1.

Response: You are right; this is the correct name for the measured quantity. Sorry, this was an annoying mistake, a misprint.

  1. Line 154. … the surface horizon (0–0.3 m)? the near-surface horizon (0–0.3 m)!

Response: We are grateful for this clarification, the text was corrected accordingly.

Author Response

Responses to comments of Reviewer 4 Report 2

Dear Reviewer 4, thank you for review our article and comments in respect of each the article text was corrected

I have re-read this manuscript, albeit with some difficulty due to the coexistence of new lines of text and rows deleted, some of which are perhaps written in Russian. However, the Authors have addressed my concerns satisfactorily. The paper is better discussed now, though the analysis of data appears rather short (but straight to the point). In conclusion, I believe this MS could be accepted for publication in Water journal almost in present form. The manuscript doesn’t exhibit evident drawbacks from the point of view of content, but some editorial changes are needed. Here below, some examples are provided.

Response: I’m sorry for difficulties to read of MS, but we have to maintain standards of Water Editorial Office, under which any revisions made to the MS should be marked up using the “Track Changes” function of MS Word. Fragments of the text in Russian remained in the manuscript by mistake, now we have deleted them.

MINOR REVISIONS

[Keywords] The keywords “Active layer” and “Large Rivers” would appear somewhat broad. I would substitute them for more specific ones. 

Response: The set of keywords has been revised, clarifications have been made, keywords that are repeated in the title of the article have been removed.

[Table 1] I have appreciated the inserting of this Table, but I would add some comments in the text. For instance, the discharge in the last column what kind of discharge is? Is it the mean annual discharge? Moreover, the unit of measurement for the discharge is wrong! It reads “m3cm-1”, but it should read “m3/s”!

Response: A short commentary to table 1 has been added to the text. Indeed, the correct name is mean annual discharge, your remark on units of measurement is also true. Sorry, this was an annoying mistake, a misprint.

[At line 175] It reads “from Beer C. et al.”, but I would write “from Beer et al.”.

Response: Thanks for the comment, the text has been edited.

[At lines from 179 to 182] I would make shorter this paragraph by writing “of the Institutes of: Permafrost, Biology, and Geography of the Siberian Branch of the USSR Academy of Sciences”. 

Response: We have shortened this paragraph without mentioning specific scientific institutions. This will allow you not to distort their names.

[Figure 2] I have also appreciated the photographs in Figure 2, but also in this case I would add some comments in the text.

Response: A short comment has been added to the text.

[Ref.# 9] It reads “upper changjiang”, but I would write “Upper Changjiang”.

Response: Thanks for the clarification, corrected.

[Ref.#19] This reference would appear somewhat complicated to read. Could the Authors carefully check?

Response: corrected
